# Role of Hydrophobic Associations in Self-Healing Hydrogels Based on Amphiphilic Polysaccharides

**DOI:** 10.3390/polym15051065

**Published:** 2023-02-21

**Authors:** Marieta Nichifor

**Affiliations:** Department of Natural Polymers, Bioactive and Biocompatible Materials, “Petru Poni” Institute of Macromolecular Chemistry, Aleea Grigore Ghica Voda 41A, 700487 Iasi, Romania; nichifor@icmpp.ro

**Keywords:** self-healing, amphiphilic polysaccharides, hydrophobic associations

## Abstract

Self-healing hydrogels have the ability to recover their original properties after the action of an external stress, due to presence in their structure of reversible chemical or physical cross-links. The physical cross-links lead to supramolecular hydrogels stabilized by hydrogen bonds, hydrophobic associations, electrostatic interactions, or host-guest interactions. Hydrophobic associations of amphiphilic polymers can provide self-healing hydrogels with good mechanical properties, and can also add more functionalities to these hydrogels by creating hydrophobic microdomains inside the hydrogels. This review highlights the main general advantages brought by hydrophobic associations in the design of self-healing hydrogels, with a focus on hydrogels based on biocompatible and biodegradable amphiphilic polysaccharides.

## 1. Introduction

Autonomous self-healing, one of the most important functions of skin, bones, and wood, is well known and is due to reversible bonds that dissipate an applied energy and prevent molecular skeleton breakage [1]. Similar to these biological materials, self-healing hydrogels were designed to recover their original properties, shape, and/or integrity after internal or external damage. They add significant improvement to mechanically unstable traditional hydrogels, due to their longer lifetime, improved safety and reduced replacement costs [2].

The healing process should occur in the absence of a healing agent and in the desired scale of time. Such hydrogels have a wide area of applications as drug delivery systems [3], wound dressings [4], cartilage and tissue engineering scaffolds [5,6], shape memory materials [7], biosensors [8], or for enhanced oil recovery [9]. For example, injectable hydrogels undergoing sol-gel transition are able to fill irregular wounds with minimally invasive implantation [10].

The self-healing mechanism is due to the presence of reversible chemical or physical cross-links with infinite lifetime inside the network [5,6,11,12]. Covalent reversible bonds such as imines, acylhydrazones, boronate esters, disulfide bonds, or bonds from Diels–Alder reactions are often used for self-healing hydrogel preparation. Dynamic physical cross-links based on electrostatic attractions, hydrogen bonds, host–guest interactions, or hydrophobic associations, give rise to supramolecular assemblies, combining the advantages of hydrogels and supramolecular chemistry [5].

Hydrogels with dynamic covalent bonds display good mechanical properties, but bonds’ reversibility is sometimes related to elevated temperatures, low pH, or presence of UV light, what could limit their applications [5]. Physical cross-linked hydrogels have the advantage of their simple preparation, without chemical cross-linkers with potential toxicity, but their applications in vivo can be limited due to poor mechanical stability. However, there are many reports on physically cross-linked hydrogels with outstanding mechanical and self-healing properties for biomedical or industrial applications [13,14,15,16,17].

Hydrophobic associations are involved in the formation of biological structures, such as micelles, membranes, or tertiary structure of proteins, and have a significant contribution to a proper adjustment of many living functions [18,19]. Hydrophobic associations are specific to amphiphilic molecules carrying both hydrophilic and hydrophobic segments, and their occurrence in aqueous media is an entropy-driven spontaneous process required to reduce or eliminate the hydrophobe’s contact with water. There is a permanent equilibrium between free and associated hydrophobes; therefore, when an external force is applied, the energy is dissipated inside the gel due to reversible dissociation, preventing the gel fracturing and endowing it with self-healing properties [20]. The strength of these gels can be modulated by varying the relative amounts of hydrophobic and hydrophilic segments and the type and size of hydrophobe [13,20]. Hydrophobic associations are generally considered as weak [6], but they are stronger than hydrogen bonds and van der Waals interactions [18], and are active at a longer distance (100–150 Å) than ionic and hydrogen bonds [21]. Moreover, hydrophobic associations can be enhanced by increasing temperature [18] or medium ionic strength [22]. Therefore, hydrophobic associations of amphiphilic polymers may generate more versatile self-healing hydrogels with well-controlled properties.

Most of the self-healing hydrogels making use of hydrophobic associations are based on synthetic polymers and imply the copolymerization of a hydrophilic monomer (usually acrylamide or acrylic acid) and a hydrophobic monomer (for example stearyl methacrylate) in the presence or absence of surfactant micelles [1,23,24,25].

Polysaccharides are natural biocompatible and biodegradable polymers, which are extensively used for the design of self-healing hydrogels, especially for biomedical applications. The preparation of polysaccharide self-healing hydrogels with contribution of either chemical or physical dynamic cross-links, as well as combination of these methods, together with their performances and applications, are very well described in recent excellent reviews [2,26,27,28,29,30]. However, the discussion about hydrogels based mainly on hydrophobic dynamic cross-links is scarce. The aim of the present review is to highlight the role of hydrophobic associations of amphiphilic polysaccharide in the performances of self-healing hydrogels prepared from them, and their potential for preparation of materials with improved properties. The direct implication of these polymers in hydrogel strength, injectability, and recovery, their contribution to improvement of hydrogel performances such as a better control of drug delivery, or to addition of new functionalities such as hemostatic, antimicrobial, and adhesive properties, will be reviewed.

## 2. Amphiphilic Polysaccharides and Their Characteristics

Amphiphilic polymers with hydrophilic and hydrophobic segments are able to self-assemble in aqueous media by hydrophobic associations with formation of aggregates of different morphologies (spherical or worm-like micelles, vesicles, hydrogels), all characterized by inner hydrophobic microdomains surrounded by hydrophilic chains. The self-assembly process is spontaneous and occurs at a concentration called critical aggregation concentration (CAC), similar to the critical micelle concentration of low-molecular amphiphiles (surfactants). The amphiphilic polymer self-assembled structures exhibit thermodynamic stability, specific rheological characteristics, surface activity, and emulsification ability [31]. Various aggregates can be obtained for different ratios between hydrophilic and hydrophobic segments and different relative positions of these segments. Synthetic amphiphilic polymers are usually copolymers of hydrophobic and hydrophilic monomers, with random or statistical distribution, or homopolymers of amphiphilic monomers. Hydrophilic polymers such as polysaccharides can be made amphiphilic by the chemical attachment of hydrophobes either as side chains (comb-like) or as end chains (block-like). These classes can be further divided according to the hydrophobes’ length: short chains (hydrophobically modified or end-modified polysaccharides) or long (polymeric) chains (grafted or block copolymers). In the case of comb-like amphiphilic polysaccharides, self-assembly takes place by intra- or intermolecular hydrophobic associations, and the ratio of hydrophilic/hydrophobic segments (degree of substitution with hydrophobic segment, DS) dictates the preponderant type of association. Usually, high DSs lead to preponderant intra-molecular associations, with formation of small nanoparticles from one single folded chain (unimers), and low DSs favor inter-molecular associations with creation of a network of physically cross-linked polymer chains via hydrophobically associated microregions. There are also intermediate cases, when both association types are present, leading to large multicomponent aggregates containing many hydrophobic microdomains [32,33]. The DS limits for each of these association cases depend on polymer flexibility, type, and length of the hydrophobes, presence of ionic groups, and polymer concentration [34].

All polysaccharide chains contain a large number of hydroxyl groups; therefore, most of them are highly hydrophilic. However, it is possible even for these chains to display some hydrophobicity, due to glycosidic ring conformation, epimeric structure, stereochemistry of the glycosidic linkages (similar to that of cyclodextrins, the cyclic oligosaccharides), or presence of hydrophobic branches and entanglements. For example, Balasubramanian et al. [35] demonstrated that linear dextrin with a *α*-1a,4e-D-glucopyranose chain has a kind of facial amphiphilicity, as all its hydroxyl groups and hydrogen atoms are arranged on different sides of the chain. The amphiphilic character was proved by the ability of this polysaccharide to bind fluorescent probes and enhance their emission intensity, and also to increase water solubility of lipophilic compounds. Other polysaccharides, like dextran and cellulose with *α*-1,6- and β-1,4-D-glucose chains, respectively, are only hydrophilic.

Hyperbranched polysaccharides can also display amphiphilic character. Lignosus rhinocerotis (LRh) has a β-(1→3)-D-glucan backbone with a hyperbranched side chain attached to every three units. The side chain contains three β-(1→6)-D-glucose units and two terminal β-(1→3)-D-glucose units. This polysaccharide was able to self-assemble with formation of hydrophobic microdomains, detected by pyrene fluorescence assay. The CAC was ~2.5 mg/mL, very close to the critical overlapping concentration (2.65 mg/mL) determined by viscosity measurements [36].

Gum arabic is another natural polysaccharide with amphiphilic properties, assigned to a small amount of hydrophobic protein covalently linked with highly branched polysaccharide. Due to its good surface activity, gum arabic is used as an emulsifier in the food industry [37].

Nevertheless, most of the polysaccharides need to be chemically modified by attachment of hydrophobic chains of different length and chemical structure. Modification with low-molecular hydrophobes (alkyl, alkenyl, aromatic compounds, bile acids) takes advantage of the presence of numerous hydroxyl, amine, or carboxyl functional groups, and coupling is realized by ester, ether, amide, carbamate, or urea linkages. Attachment of polymer chains is usually performed by “grafting to” and “grafting from” methods [38]. “Grafting to” is a reaction among functional groups of the two preformed polymers and derivatization pathways are similar to those used for binding low-molecular hydrophobes. This procedure is efficient only for polymer with a low degree of polymerization. In the “grafting from” method, an initiation center is first created at polysaccharide functional groups, from which the polymerization propagation of the chosen monomer will start. This method allows a better control of the side chain length and dispersity. Chemical modification of polysaccharides, as well as synthesis of hydrophobically modified polysaccharides, have been the subject of many reviews and will not be discussed here [39,40,41,42,43,44]. It is worth mentioning the reaction between amine and aldehyde groups with formation of imine bonds, often used for the preparation of self-healing hydrogel based on polysaccharides, especially when chitosan is one of the components. Imine groups can be formed by the reaction of chitosan NH_2_ with aldehyde groups of low-molecular or macromolecular compounds (including hydrophobic molecules), as well as with aldehyde groups of other polysaccharides, obtained by their oxidation in the presence of NaIO_4_ [45].

## 3. Evaluation of Self-Healing Hydrogel Performance

There are many experimental techniques used to assess the main properties required for a self-healing hydrogel, and they are well presented in several comprehensive reviews [11,46,47,48,49]. The measurements are not fully standardized; therefore, the comparison of the results of different studies may be difficult [46]. Nevertheless, useful information about performances of the hydrogels described in the next sections, such as viscoelasticity, strength, injectability, and ability to recover the original properties after damage by an external force, could be obtained by rheological and mechanical measurements.

Hydrogels’ viscoelasticity and shear-thinning ability can be well demonstrated by rheological measurements [46,47]. A frequency sweep experiment (usually up to 100 rad/s) shows the relationship between storage modulus (G′, a measure of the gel elasticity and strength) and loss modulus (G″). The gel elasticity and strength are given by the value of the storage modulus G′ and the phase angle, δ, (*tan* δ = G″/G′). Higher G′ and lower *tan* δ will indicate an elastic and strong gel. Strain-dependent oscillatory rheology shows the strain range over which the viscoelastic behavior is maintained, and the decrease of G′ will indicate the network destruction under stress. The strain corresponding to the G′ = G″ is defined as yield stress (γ), the minimum force required for a gel to flow. A high-yield stress is a measure of the gel strength. Injectability can be demonstrated either by the capacity of strain-thinning (G′ and G″ moduli decrease with the increase of strain) or shear-thinning (decrease in viscosity with increasing shear rate), both reflecting the transient state of dynamic cross-links. A moderate value of yield stress is appropriate for gels designed as injectable materials, and its required value and the corresponding viscosity depend on the device used (syringe or catheter), as well as the length and diameter of the syringe needle [48].

Mechanical properties of a hydrogel can be evaluated by tensile strength, elongation, or compressive strength values before break. Tensile strength is the resistance of a material to breaking under tension. The optimal values depend on the intended application, and a tensile stress oscillating between 0.1 and 1 MPa is commonly considered to be strong enough for satisfying hydrogel general applications [11].

The self-healing properties of hydrogels, i.e., the reversibility of their dynamic interactions, were evaluated by a different procedure. Reversible sol-gel transitions under the application of a stress, a property required for local application by injection, can be evaluated by rheological measurements, which will indicate the recovery of initial values of G′ and G″ after alternate step stress and the time necessary for this recovery process [11]. Remolding (or reformation) of a hydrogel implies the cracks/cuts’ repair and was usually observed visually, by monitoring the time required for two cut pieces to stick together (gel block fusion test [46]) without external intervention. Optical microscopy can give more information on cracks’ closure [49]. The resistance to manipulation of the remolded material, as well as its mechanical properties, have been sometimes measured [46]. A shape recovery test was reported in several cases for materials designed as wound dressing, the purpose being the finding of the time required for a gel to recover its initial dimension in the swollen state, as well as the extent of this dimension recovery [50].

## 4. Amphiphilic Polysaccharides as Active Components in Self-Healing Process

### 4.1. Pure Hydrophobic Association Hydrogels

#### 4.1.1. Hydrogels Based Only on Amphiphilic Polysaccharides

Supramolecular assembly is a characteristic of many unmodified polysaccharides; for example, polysaccharides from plant cells or different kinds of cellulose, with shear-thinning and self-healing properties [51]. In most cases, self-assembly is due mainly to hydrogen bonds, which will be present in all the systems containing polysaccharides.

There is only one case of an unmodified polysaccharide with intrinsic amphiphilicity whose self-healing behavior was evaluated. The rheological behavior of a LRh/water system highlighted a shear-thinning phenomenon depending on concentration, and a complete recovery to the initial state after the strain removal when concentration was >15 mg/mL [52]. Temperature had a negative effect on self-healing as it induces irreversible changes, but a short-term (10 min) ultrasound treatment promoted LRh hydrophobic aggregation, proved by the decrease of CAC and the increase in hydrophobic microdomain number, and resulting in the increase of the apparent viscosity, storage modulus, loss modulus, and the recovery level of the LRh/water system [53]. Regarding polysaccharides with chemically induced amphiphilicity, rheological studies on aqueous systems containing hydrophobically modified alginic acid [54] chitosan [55], gellan gum [56], hyaluronic acid [57], pullulan [58], or cellulose [59] have revealed their ability to form strong hydrogels with shear-thinning properties, indicating their potential use as injectable materials, but the self-healing properties were only suggested, not investigated.

However, there are some reports dealing with self-healing hydrogels built only from amphiphilic polysaccharides. For example, Fredrick et al. [60] prepared a nontoxic and hemocompatible material by attaching dioctyl amine (DS = 33%) to carboxymethyl cellulose (CMC), which formed a gel at concentration ≥10 wt% in water. The strain-sweep test showed that the value of storage modulus G′ of this gel was higher than that of chemically cross-linked CMC, at the same frequencies of the strain sweep, the shear-thinning starts to occur at a shear strain higher than 10% and a complete gel-to-liquid transformation was observed at a strain >238%. A step-strain test indicated a complete and fast recovery of the gel properties after the strain removal, but only when a strain ≤200% was applied. At higher strain, the self-healing process was much slower.

An interesting amphiphilic polysaccharide was prepared by chemical attachment of ferrocene (Fc), a hydrophobic biocompatible molecule, to chitosan (Ch), via amide groups [61]. Fc-Ch conjugate (containing 8.2 mol% Fc) formed a self-standing hydrogel at 1 wt% in acetic acid due to the association of ferrocene groups, which act as cross-linking points. Rheological measurements showed clear viscoelastic properties, as G′ was one order of magnitude higher than G″ over all frequency range, shear-thinning at an yield stress of 100% and a rapid recovery of the initial moduli values when step strain was applied by changing strain amplitude from 500 to 1% (at 1.6 rad/s). This gel had also remolding properties. A gel in the form of a disc cut into two pieces completely reformed 4 h after the two pieces were rejoined. The reformed disc had the same G′ and G″ as the original disc and could be stretched to almost twice its original diameter. Due to intrinsic properties of the two components, chitosan and ferrocene, Fc-Ch gel showed also pH, redox, and ion responsiveness, which favored controlled release of doxorubicin.

Besides attachment of small hydrophobic molecules, amphiphilic polysaccharides were obtained by grafting hydrophobic polymers. For example, injectable hydrogels were synthesized by attaching polyurethane (PU), by a “grafting-to” procedure, to chitosan [62] or dextrin (Dx) [63], via urethane or urea chemical bonds obtained by reaction between PU end isocyanate groups and NH_2_ or OH groups of the polysaccharide. The amphiphilic character of graft copolymers was demonstrated by their having a higher contact angle than that found for pure polysaccharide. Contrary to chitosan, which instantaneously formed a gel when dissolved in 0.1 M acetic acid, Ch-*g*-PU gelation was a longer process, allowing its use as an injectable material. The hydrogel strength and viscosity were both one order of magnitude higher than those of pure chitosan. The gel formed after in vivo subcutaneously injection was stable in time and did not induce inflammation or damage. The sustained delivery of an anticancerous drug (doxorubicin hydrochloride) and the increased in vitro efficacy of the encapsulated drug showed the potential of this new polymer for drug local delivery. Similarly, Dx-*g*-PU gel showed a high mechanical strength, with a Young modulus of 3 and 5 MPa, for lower (15%) and higher (30%) grafting density, respectively, as well as high elongation at break. Dx-*g*-PU had to be mixed with 10% methyl cellulose (MC) as a gelation agent in order to obtain an injectable material. The different gelation behavior of Ch-*g*-PU and Dx-*g*-PU can arise from different diols used for PU preparation, i.e., poly(ε-caprolactone) chains connected by an ether bridge and poly(tetramethylene glycol) in case of Ch-*g*-PU and Dx-*g*-PU, respectively, leading to a more hydrophobic Dx-*g*-PU. Dx-*g*-PU gel injected subcutaneously could be placed just below the tumor site and release the drug in a sustained manner, exclusively to the tumor site [63].

#### 4.1.2. Hydrophobic Associations Enhanced by Surfactant Micelles, Nanoparticles, or Vesicles

The concentrations at which amphiphilic polysaccharide can form viscoelastic gels via intermolecular hydrophobic associations are usually high and depend on polysaccharide chain flexibility and hydrophobe length and nature. Additional hydrophobic interaction with self-assembled materials such as micelles, vesicles, or nanoparticles can lead to much stronger gels at lower polymer concentrations.

Interaction of amphiphilic polymers with surfactants has been extensively studied and showed that, at a certain ratio of hydrophobes to surfactant, gelation can occur [34]. This gelation process was applied in development of self-healing hydrogels prepared by micellar copolymerization of a hydrophilic and a hydrophobic monomer [1,25,64], where surfactant micelles were introduced for the solubilization of the water insoluble hydrophobic monomer.

The mixed micelles formed between hydrophobic monomer and surfactant act as cross-linking points connecting the hydrophilic segments of the resulting polymer (Figure 1) [65]. The procedure was extended to polysaccharides, which were grafted by the “grafted from” method, with mixtures of hydrophilic and hydrophobic monomers in the presence of a surfactant as a hydrophobic monomer solubilizer. In such a system, Duan et al. [66] grafted cellulose nanowhiskers (CNW) with a copolymer acrylamide (Am)/alkyl methacrylate (CnMeMA) with different alkyl chain length (Cn = C12, C13, C18), in the presence of sodium dodecyl sulfate (SDS) micelles. Octadecyl segments provided the cross-linking points by hydrophobic associations occurring both inside and outside SDS micelles. The resulting hydrogel had outstanding mechanical properties, which could be optimized by the appropriate choice of the gel composition. CNW contributed to both gel reinforcement and swelling capacity. The best results were obtained when the most hydrophobic monomer C18MeMA was used for gel preparation, showing the dominant role of physical cross-links formed by hydrophobic association. After cutting a gel piece in two parts, a remolding process took place, which was completed after 60 min. The material also displayed excellent mechanical properties: 2500% extensibility, and high tensile (1.338 MPa) and compressive (2.835 MPa) strength. However, the self-healing property was lost after SDS extraction with water, a phenomenon already highlighted by Tuncaboylu et al. [67] for the gels obtained from Am and alkyl acrylates and explained by the local solubilization of hydrophobes inside SDS micelles. In the absence of surfactants, the hydrophobic associations between long alkyl chains become very strong and long-lived, affecting their reversability. In a similar approach, Duan et al. [68] grafted Am-*co*-C18MeMA to galactomannan in aqueous solution in the presence of SDS micelles. The amount of C18MeMA had a remarkable influence on the mechanical properties of the hydrogels. An optimum content (8–10% related to Am) led to the best results in terms of strength and compression resistance. Two cut material pieces were autonomously reconnected in seconds at room temperature, and the cut surface trace disappeared in 10 h. However, there was no information on whether SDS was removed from the gel before mechanical and self-healing tests.

Due to potential cytotoxicity of SDS (and other surfactants), efforts have been made to replace it with biocompatible amphiphilic molecules that can form stable hydrophobic associations. For example, Meng et al. [69] used silk fibroin, a natural amphiphilic compound, which consists of both hydrophobic crystalline regions and hydrophilic amorphous regions and is able to self-assemble in micelles. They obtained hydrogels from alginic acid sodium salt (SA) grafted with stearyl methacrylate solubilized in silk fibroin micelles. The occurrence of hydrophobic associations between grafted hydrophobes and silk fibroin was revealed by FTIR spectroscopy. In this case, the gel strength was further improved by SA cross-linking in the presence of CaCO_3_. Perhaps the gelation of the system Dx-*g*-PU obtained only after addition of MC [63] was the result of a similar process, as amphiphilic MC may play the role of a surfactant which partially disrupts the intramolecular strong hydrophobic associations between grafted PU chains. It is worth mentioning here the use of amphiphilic gum arabic as an additional surfactant besides SDS in preparation of a self-healing hydrogel based on Am and lauryl methacrylate [70]. The combined surfactants had a synergistic effect on hydrophobic segment stabilization, resulting in an increase of yield stress from several hundred pascals to 1 MPa in the absence and in the presence of gum arabic, respectively.

Another approach to enhance hydrophobic associations in physically cross-linked self-healing hydrogels is the interaction between amphiphilic polymers and nanoparticles (NPs’) hydrophobic core or vesicle bilayer. In contrast to the case of pure amphiphilic polymers or amphiphilic polymers/surfactant systems, where cross-links are provided by intermolecular hydrophobic associations between the pendent hydrophobes, the amphiphilic polymer has a few points of interaction with NPs/vesicles and acts mainly as a bridge between these particles, resulting in supramolecular networks in the form of stiff gels [71,72]. Such networks based on reversible hydrophobic associations between amphiphilic polymers and NPs could display shear-thinning and self-healing properties, as shown for systems containing copolymer Am-CnMeMA and latexes [1].

Based on this concept, Langer and Appel research groups [72,73,74,75,76,77] performed an extensive work on injectable and self-healing hydrogels obtained by interaction between hydrophobically modified hydroxypropyl methyl cellulose (HPMC) and different core-shell NPs. HPMC was modified by chemical attachment of alkyl chains (C6 or C12) or adamantyl groups [72]. The tests showed that stronger polymer-NP gels were obtained by increasing the hydrophobicity and density of the substituent attached to HPMC, an indication of the important role played by hydrophobic associations in gel formation [72,73,74]. The optimum substitution degree with C12 side chains was found to be 2 mol%, which allowed a good enough interaction of polymer-NPs, but avoided the wrapping of the polymer around a single Np and favored polymer bridging of multiple NPs with formation of the hydrogel (Figure 2A). The bridging depended on the NPs’ size, which should fit to polymer persistence length (*l*_p_). As HPMC’ *l*_p_ ~ 90 nm, NPs with a hydrodynamic diameter ~50–80 nm were chosen for gel formation [72,74]. NPs obtained from carboxylated polystyrene [72], polystyrene [74], and biodegradable copolymers of polyethylene glycol-*b*-poly(lactide) (PEG-*b*-PLA) [73,74,75,76,77] were used in these experiments. Study of the interaction between polystyrene NPs and modified HPMC offered information about the main factors of influence on gel properties (concentration of each component, total occupied volume fraction, size of NPs) [72,74], but a lot of attention was dedicated to the HPMC-C12/PEG-*b*-PLA system. PEG-*b*-PLA can self-assemble to form spherical nanoparticles with shell-core morphology [75]. The use of this biodegradable and biocompatible polymer opened up the possibility to design hydrogels able to encapsulate both hydrophobic drugs inside a block-copolymer core and hydrophilic compounds in the aqueous gel part. Detailed rheological evaluations highlighted the influence of component chemical structure, their relative weight ratio, and volume fractions on viscoelasticity, shear-thinning, and self-healing behavior of hydrogels [73,75]. Use of PEG with different chain lengths (2 and 5 kDa) for the preparation of block-copolymer NPs [73] revealed a better behavior of the gel containing PEG 5kDa, in terms of moduli values, crossover frequency, and recovery time. This finding suggested a stronger interaction between HPMC-C12 and NPs having 5kDa PEG, contrary to the initial hypothesis that a shorter block of NPs’ shell will allow better access of polysaccharide side chains to the NPs’ hydrophobic core. Variation of the components’ content provided information about the contribution of each component to hydrogel properties (Figure 2B) [75]. HPMC-C12 contributes to viscosity increase and its excess reduces shear-thinning behavior, while PEG-*b*-PLA’ NPs determine solid-like properties of hydrogels. An optimal polymer/NPs ratio of 0.1–1 was found, with best performances observed for a ratio close to 0.5. Further, Appel and coworkers studied the relation between hydrogel rheological properties and depot formation and its resistance after subcutaneous administration to mice [76], an important aspect related to application of these systems as efficient drug delivery platforms, since a depot’s low mechanical stability could lead to inflammation and an undesired drug burst release. Their results showed that materials able to provide extended local delivery of therapeutic agents should display yield stresses >25 Pa and a reduced creep in order to achieve depot robustness and longer depot persistence, respectively. HPMC-C12/PEG-*b*-PLA hydrogels have been successfully tested in vivo, after subcutaneous injections on animal models, for complex delivery of biologically active substances. Good results were obtained for co-delivery of both hydrophilic and hydrophobic drugs [72], as vaccine platforms [78,79] when both antigen and its adjuvant could be delivered with similar release rate, despite their differences in molar mass, for in vivo immunomodulation [73] or cell delivery [80]. When hyaluronic acid carrying alkyl groups (C8, C12, C14) (Hy-Cn) was used instead of the cellulose derivative, the system Hy-Cn/PEG-*b*-PLA provided hydrogels with variable stiffness, tuned by variation of hyaluronic acid DS, molecular weight, and content. These hydrogels with shear-thinning and self-healing behavior could be injected through high-gauge needles and long, narrow catheters. Hydrogels preserved the biological function (regenerative and anti-inflammatory effects) of the hyaluronic acid component, enhancing the activity of these materials [81].

In the attempt to extend the applicability of their HPMC-C12/NPs hydrogel platform, Apple and co. [77] replaced PEG as a NP corona of block copolymers with somewhat more hydrophobic polymers, such as poly(isopropyl acrylamide) (PNIPAM) or poly(diethylacrylamide) (PDEAm), and obtained materials with a lower yield stress (<10 Pa), but with a high extensibility (up 2000% strain to break) and significant water content (93%), properties required for applications such as 3D printing materials, adhesives, injectable biomaterials, and in foods.

Another system based on interaction between an amphiphilic polysaccharide and NPs was proposed by Nigmatullin et al. [82], who prepared a hydrogel based on hydrophobic associations between two cellulose derivatives: HPMC and cellulose nanocrystals (CNC) modified with 4 mol% octyl groups (CNC-C8). Both derivatives have amphiphilic character, with surface activity and measurable *CAC*. In this system, HPMC is the soluble polymer and CNC-C8 represents NPs. The occurrence of hydrophobic association between the two polymers was demonstrated by Raman spectroscopy and had as a result the formation of robust hydrogels with higher storage modulus and two orders of magnitude higher viscosity than HPMC or HPMC/CNC systems, where hydrogen bonds are dominant. The strain- and shear-thinning were observed for all the systems, but HPMC/CNC-C8 hydrogel showed a slower recovery of its original rheological properties (90% in 3 min, 100% in 15 min).

Similar to interaction between hydrophobically modified polysaccharides and nanoparticles, the hydrophobic segments of these polymers can interact with vesicle bilayers. Lee et al. [71] performed a detailed study of the rheological behavior in systems containing Ch-C12 (DS = 2.5 mol%) and a mixture of cetyl trimethylammonium tosylate (CTAT)/sodium dodecyl benzene sulfonate (SDBS), which could aggregate as vesicles or worm-like micelles, when the ratio of CTAT/SDBS was 70/31 and 100/0, respectively. At a polymer concentration above its overlapping concentration, elastic gel formation in the presence of vesicles was observed, without vesicle disruption. The gels had shear-thinning behavior and their strength increased with an increasing amount of both components. It was suggested that several hydrophobes of a single polymer chain were embedded in different vesicle bilayers, resulting in a network where vesicles play the role of the multifunctional cross-linkers (Figure 3). Interesting, Ch-C12/worm-like micelle systems formed only viscous transient networks. The difference was assigned to the much lower stability of worm-like micelles.

Hao et al. [83] applied the same approach by mixing Ch-C12 (DS = 2.74 mol%) with vesicles obtained from a mixture of dodecyltrimethylammonium bromide (DTAB)/5-methyl salicylic acid (5 mS). The resulting hydrogel exhibited excellent stability at mild temperature and the strength of the gels could be tuned by the content in 5 mS. Rheological measurements performed at 20 °C revealed the formation of strong gels with a yield stress of 186% and a rapid (10 s) and complete recovery of initial moduli values after stress removal. Moreover, the gels showed reversible thermoresponsiveness, due to temperature induced vesicles-to-micelles transitions of the DTAB/5mS mixture, leading to disruption of the hydrophobic interactions between Ch-C12 and surfactant short-lived worm-like micelles. Consequently, polymer/vesicles mixtures displayed a gel-like behavior at low temperature (15 °C), were viscoelastic gels at 20 °C and solution at 50 °C. In case of the mixture Ch-C12 (10 mol%)/monoolein-sodium oleate vesicles [84], the shear rate dependence highlighted a more complex profile, characterized by a Newtonian plateau followed by intensive shear-thinning.

Liposomes were also used to design hydrogels based on hydrophobic associations [85,86]. Hydrogels prepared by Correa et al. [85] from HPMC-C12 and liposomes were stiff gels when liposome concentration was >1%, with shear-thinning and self-healing behavior. The change of liposome surface chemistry, by using either anionic or cationic lipids, or by chemical modification with affinity molecules or PEG, did not affect hydrogel mechanical properties, but opened opportunities to manipulate the release rate of proteins, irrespective of their size, an issue for application in immuno-engineering and tissue regeneration, where the results are influenced by the sequence of several drugs’ release.

Hajizaki et al. [86] used HPMC with octadecyl pendent groups and liposomes prepared from hydrogenated soy phosphatidylcholine and dicetyl phosphate. Rheological measurement confirmed the viscoelastic gel formation, with reversible shear-thinning properties.

### 4.2. Self-Healing Hydrogels with Multiple Interactions

A single type of reversible cross-linking rarely leads to the required mechanical properties and self-healing ability, therefore, combination of two or more physical and/or chemical dynamic bonds is often used in the design of hydrogels with improved properties and more appropriate to a certain application [87]. There are several examples of amphiphilic polysaccharides used as components of dual or multiple cross-linked networks.

#### 4.2.1. Hydrophobic Associations and Other Physical Cross-Linking

As already mentioned at Section 4.1.2, Meng et al. [69] prepared a hydrogel based on alginic acid grafted with stearyl methacrylate solubilized in silk fibroin micelles. In order to increase the network stability, a supplementary gel cross-linking was added, by ionic interaction between alginic acid and CaCO_3_. The role of hydrophobic association between hydrophobic grafts was decisive for the properties of this double physically cross-linked hydrogel. At an optimum hydrophobic graft content, the values measured for compression strength (17.1 kPa), Young modulus (0.37 kPa), storage modulus G′ (7 kPa), and yield stress (109%) were much higher than the corresponding values measured for the gel prepared in the absence of the hydrophobic monomer (5.3 kPa, 0.16 kPa, 0.35 kPa, 1.59%, respectively). The presence of hydrophobic associations had a decisive contribution to a fast recovery of the elastic properties after strain removal, as well as the rapid reshaping of the crack created on the gel surface, showing the self-healing properties of these gels.

Liu et al. [88] proposed a multi-cross-linked system based on two hydrophobically modified and oppositely charged polymers, where the complex hydrophobic associations are supplemented by ionic and hydrogen bonds. The mechanical properties of the hydrogel, containing the amphiphilic cationic poly(N,N-dimethylacrylamide-co-N-dimethyl-N-stearyl aminopropylacrylamide) (PAM-18), cross-linked by hydrophobic associations between C18 groups, was significantly improved by addition of a second amphiphilic anionic polymer, CNC-C8, which introduced extra hydrophobic associations between C18 and the C8 alkyl chain, as well as ionic interactions between amine groups of PAM-18 and anionic groups (SO_3_ and COO^−^) of CNC-8. All these physical cross-links contributed to mechanical properties of the final gels, and the maximum tensile strength, obtained for the optimal compositions of the gels, increased with an increasing number of interactions, and was 130, 238, and 331 kPa for PAm-C18, PAm-C18-CNC, and PAmC18-CNC-C8, respectively. Elongation increased in the same order. Hydrogels resisted deformation and could be repaired by addition of THF between two cut pieces.

#### 4.2.2. Hydrophobic Associations and Chemical Cross-Linking

Cai et al. [89] succeeded in improving the mechanical properties of an amphiphilic grafted polymer by chemical cross-linking via irreversible covalent bonds. For this purpose, carboxymethyl dextran (CMD) was reacted with the end OH groups of short chain poly(ε-caprolactone) (PCL), and the amphiphilic grafted polymer was further cross-linked with PEG, via ester groups. The resulting hydrogels had good resistance to compression and preserved elasticity and self-recovery endowed by dynamic hydrophobic cross-links. In vitro and in vivo tests highlighted the hydrogel potential for bone tissue engineering.

Another approach for improving hydrogel properties was to bind an amphiphilic cross-linker to glycol chitosan via dynamic imine groups. Cai et al. [90] synthesized a multifunctional amphiphilic cross-linker carrying benzaldehyde groups able to react with chitosan amino groups. The cross-linker was an amphiphilic ABA telechelic copolymer of N,N-dimethyl acrylamide and N-hydroxyethyl acrylamide (HEAm), with PEG as a B block, and benzaldehyde groups introduced by modification of the HEAm monomer with formyl benzoic acid. The gelation time, gel strength, viscoelasticity, and self-healing were tuned by the hydrophobic benzaldehyde groups/ABA copolymer chain and by variation of the molar ratio between cross-linker CHO groups and chitosan NH_2_ groups. The remolding of cut pieces took place after a long time (24 h), but required only 5 h when a very small amount of NaOH solution was added. The gel mechanical and self-healing properties could be improved by addition of graphene oxide, due to additional hydrogen bonds. The hydrogel was supposed to act as a good depot for hydrophobic drugs, which could be easily entrapped in cross-linker micelles.

### 4.3. Temperature-Induced Hydrophobic Associations

The systems presented until now are characterized by an autonomous self-healing process; i.e., they can reversibly change their properties in the absence of healing additives or external stimuli. Gels that undergo stimuli-dependent sol-gel transition have recently attracted attention as promising biomaterials for injectable materials, and among them, the thermoresponsive gels play an important role. There are several synthetic polymers that show lower critical solution temperature (LCST), such as PNIPAM and poly(propylene oxide)-*b*-poly(ethylene oxide)-*b*-poly(propylene oxide) triblock copolymers (poloxamers or pluronics). Above LCST, these polymers become water insoluble due to the destruction of hydrogen bonds between water and different groups along the polymer chains, leading to enhanced hydrophobic interactions and polymer separation as precipitate or hydrogel. This process is reversible, so it can be used for temperature-dependent self-healing. Unmodified polysaccharides are not thermosensitive, apart from a few of them (agarose, carrageenans, gellan gum) characterized by upper critical solution temperature, but this property has limited biomedical applicability [91]. Polysaccharides grafted with polymers that display LCST have been extensively used for development of new hydrogels with thermosensitive and self-healing properties, and they are well presented in a review [91] and some more recent scientific reports [92,93,94,95,96,97]. The present review will focus on several amphiphilic polysaccharide derivatives with their own thermogelling properties, obtained by attachment of low-molecular compounds to polysaccharide functional groups.

Some water-soluble cellulose derivatives, MC, hydroxypropyl cellulose, and HPMC, are thermosensitive, but their transition sol-gel is a step-wise process and strongly dependent on concentration [98,99]. MC was found to display gel like-behavior, shear thinning and gel properties recovery even at 23 °C, when its concentration was ≥8 wt%, but the gel strength sharply increased at temperature above 37 °C, as G′ was two orders of magnitude higher [100]. Due to poor mechanical stability of its gels, MC was used in combination with other polymers, to add supplementary physical and/or chemical dynamic bonds. For example, oxidized MC (Ox-MC) was mixed with chitosan oligomer to form a hydrogel dual-cross-linked via imine groups and thermoreversible hydrophobic associations [101]. Gelation temperature and gelation time could be adjusted by the weight ratio of the two polymers. In comparison to MC or MC/chitosan, the gel Ox-MC/chitosan (at optimal ratio 4/1) prepared at 37 °C showed a significant improvement of compressive stress and gel strength (evaluated by G′ value), and displayed better self-healing behavior (Figure 4). Two model drugs, (adenosine and vitamin C) encapsulated into the gel, were slowly released over a period of 72 h.

In the system proposed by Nigmatullin et al. [82], both components, HPMC and CNC-C8, are amphiphiles at ambient temperature, but their hydrophobicity and self-assembly are enhanced by raising temperature in the range of 40–65 °C, as both are thermoresponsive. Consequently, the density of hydrophobic cross-links between HPMC and CNC-C8 is increased at higher temperatures, leading to stiffer networks, as evidenced by a significant enhancement of rheological properties in both oscillatory rheology and steady shear viscometry.

Chitosan modification with short alkyl substituents leads also to interesting thermoresponsive derivatives. Hydroxybutyl chitosan (HBuCh), prepared from chitosan and 1,2-epoxybutane, is commercially available and displays LCST in the range 17–29 °C, as a function of DS (1.3–1.7 mol substituent/repeating unit) [102] and concentration [103].Depending on synthesis method, the thermal transition of aqueous HBuCh solutions lead either to gels (heterogeneous medium, uneven chemical modification) or to colloidal dispersion/precipitates (homogeneous medium, even chemical modification) [104,105]. The HBuCh sol-gel transition is fast (30–60 s at 37 °C), and the gel displays a solid-like behavior [102,106]. This polymer, which preserves chitosan biological properties, could be formulated as gels, sponges, nanoparticles, fibers, films, or other complex structures and was tested, alone or in composite materials, as wound dressing, tissue engineering scaffold, cell culture, or cell and drug carrier [107]. HBuCh hydrogel stimuli response and mechanical strength could be improved by different approaches. By addition of salts, Wang et al. [108] succeeded in enhancing gel strength in the presence of 5% NaCl (G′ increased from 195 to 428 Pa) and decreased the gelation temperature. Blending of HBuCh with poly(sulfobetaine methacrylate) resulted in a fully physically cross-linked hydrogel by both thermo-induced hydrophobic associations and electrostatic interactions [109]. The hydrogel was characterized by thermosensitivity, self-healing, antibiofouling, and synergistic antibacterial activity, and could be applied in postoperative anti-adhesion and healing of infected wounds.

Another interesting thermo-gelling polymer based on chitosan was synthesized by attachment of hexanoyl groups to glycol chitosan and tested for different biomedical applications [110,111,112]. The thermo-reversible gelation of GCh-C6 results from the presence of hydrophobic hexanoyl groups that allow GCh-C6 chains to be physically cross-linked by temperature-dependent hydrophobic associations. Gelation temperature could be tuned by the DS with C6 groups and polymer concentration, and was about 30–34 °C when DS ≈ 37% and c = 3–4 wt% (Figure 5). This chitosan derivative showed mucoadhesiveness [110], a property required for the proper healing of damaged soft tissues. The biocompatible hydrogel formed after injection into the degenerative intervertebral disc tissue of an ex vivo porcine model maintained its stability for more than 28 days, highlighting the potential gel applicability in disc herniation treatment [111]. In an attempt to improve adhesive properties of GCh-C6 polymer, Park, Li et al. [112] conjugated it with gallic acid (GA), the pyrogalol moiety of which is able to form cross-links through intra/intermolecular hydrogen bonding and chemical bonds. The obtained GCh-C6-GA derivatives displayed sol-gel transition at a similar temperature (35 °C) as the starting polymer (34 °C), but gelation process was slower (200 s and 71 s, respectively) due to the presence of hydrophilic GA. The hydrogel formed at 37 °C showed viscoelastic properties, a yield stress of about 10% and self-healing ability proved both by rheological and remolding tests. GA attachment enhanced chitosan-derivative mechanical properties, due to pyrogallol moiety oxidation to o-quinone, accompanied by radical formation leading to chemical bonds and cross-links. Consequently, the compressive strength increased from 0.01 to 0.37 MPa when GA content was 0 and 10%, after 1 day under ambient conditions, and the process was time-dependent. The adhesive strength, tested on porcine skin, increased from 27 to 70 kPa for 0 and 10% GA content, and was enhanced at higher temperature. In the in vivo experiments, GC-C6 (37%)-GA(10) exhibited excellent wound healing activity in a full-thickness skin defect model in mice. Further, Lee et al. [113] blended GCh-C6 with acetylated hyaluronic acid (AcHy, DS = 27%), under the hypothesis that the hyaluronic acid addition would lead to an injectable thermogel with improved mechanical properties and cell binding affinity, due to supplementary ionic interaction between the two polysaccharides and intrinsic hyaluronic acid affinity for cells. AcHy was a better choice than Hy due to its amphiphilicity, allowing a better miscibility with GCh-C6 and better anti-inflammatory activity [114]. In contrast to GCh-C6, Ch-C6/AcHy blend underwent a thermo-irreversible sol-gel transition as a result of multiple physical cross-linking by hydrophobic associations, ionic interactions, and hydrogen bonds. The sol-gel transition was time-delayed at ambient temperature and was instantaneous at body temperature, allowing both injectability in an appropriate time interval and depot stability. The gel mechanical strength increased with time and temperature. The biocompatible and biodegradable hydrogel showed more effective cartilage formation than the original GCh.

Otto et al. [115] tried to mimic the chemical structure of PNIPAM by attachment of alkylamide groups to dextran, via a succinoyl bridge. The resulting polymers displayed thermoresponsivity in a temperature range of 5–40 °C. Transition temperature and transition time depended on alkylamide hydrophobicity and DS. No transition was observed for DS ≤ 1 (mol substituent/glucosidic repeating unit). It has to be mentioned that thermal transition led to colloidal dispersions, with particle size of 400–600 nm. No hydrogel formation could be observed, even at concentration as high as 10–40%. The new polymers were biocompatible in mice following subcutaneous and intercranial injections; thus, they have potential as injectable colloidal drug delivery systems.

## 5. Additional Role of Amphiphilic Polysaccharides as Components of Self-Healing Hydrogels

There are some self-healing hydrogel designed for local treatment, in the composition of which amphiphilic polysaccharides are included, but hydrophobic associations do not directly influence the self-healing process, instead they endow new and multifunctional properties of hydrogels; for example, they allow a better encapsulation and release of hydrophobic drugs, or offer hemostatic, antibacterial, and/or adhesive properties.

### 5.1. More Efficient Hydrophobic Drug Encapsulation

Water-insoluble hydrophobic drugs can aggregate and precipitate inside a hydrogel, resulting in an inhomogeneous system, which affects both drug release kinetics and hydrogel mechanical properties. A more efficient hydrophobic drug delivery can be achieved by integration of drug-loaded micelles into the hydrogel matrix [116], leading to multicompartment hydrogels with complex functionality [117]. Micelles formed by amphiphilic polymers are able to encapsulate hydrophobic drugs in the micelle core and release them in a controllable manner. Many amphiphilic polysaccharides have been proposed as such drug delivery systems [43,118,119], administered systemically due to their colloidal stable form. Addition of such a drug/polymer micelle component to self-healing hydrogels would allow the preparation of injectable materials with improved local drug delivery.

Chitosan chemically substituted with carboxymethyl and hexanoyl groups (CMCh-C6, DS of about 0.5 for both substituents) can aggregate in water medium with formation of nanosized micelles or nanocapsules, which efficiently encapsulate hydrophobic drugs such as doxorubicin or demethoxycurcumin [120]. Lin et al. [121] have incorporated this amphiphilic chitosan in an alginate hydrogel matrix ionically cross-linked with Ca^2+^. The concentration of CMCh-C6 was above its CAC, thus the structure of the formed composite gels was proposed to be a cross-linked alginate matrix with embedded nano-micelles (50–200 nm diameter). CMCh-C6 was neutralized to its isoelectric point (~7.5) and no aggregation or gelation was observed in the mixture of the two polysaccharides before CaCl_2_ addition. The composite gel, containing glycerol as a connector via hydrogen bonds between non-interacting components, was viscoelastic (tan δ = 0.2), underwent gel-sol transition at y = 17.8% and recovered very fast its initial gel state. When CMCh-C6 micelles encapsulated with retinoic acid were used for hydrogel preparation, the drug release from the composite gel was very slow, which recommended the material as a long-term local drug delivery system when applied as an injectable depot or for dermal applications. In a similar approach, Lu et al. [122] obtained an injectable hydrogel from a mixture of CMCh-C6/low-molecular hyaluronic acid, but in this case the gelation, which took place at pH < 7, was the result of both electrostatic interaction between the chitosan micelles’ surface and hyaluronic acid, and micelle bridging by hyaluronic acid. The gel displayed shape recovery and adhesive properties. Berberine, a drug with antibiotic and anti-inflammatory activities, could be encapsulated in the gel and released in a sustained and pH-dependent manner.

Another way to obtain an injectable hydrogel with improved mechanical strength and controlled release was reported by Yan et al. [123], who used an amphiphilic maltodextrin derivative, obtained by conjugation of palmitic acid to polysaccharide, (MDx-C16), as both carrier for hydrophobic drug simvastatin (SIM) and as hydrogel cross-linker. For this purpose, MDx-C16 was oxidized to generate aldehyde groups on MDx chains and MDx-C16-CHO loaded with SIM was mixed with carboxymethyl chitosan, resulting in a hydrogel cross-linked via imine bonds. The gelation time ranged from 40 to 80 s and depended on the oxidation degree of MDx-C16-CHO. The hydrogels displayed improved properties compared with a similar system without imine bonds: 2.0–2.4 times higher G′ values, good compression resistance and elasticity, and shape recovery after compression removal. Composite hydrogels obtained from SIM-loaded dextrin micelles reduced drug biotoxicity, provided a prolonged drug release without burst effect, and improved the osteogenic differentiation of MC3T3-E1 cells.

### 5.2. Hemostatic, Antibacterial, and/or Adhesive Properties

When an amphiphilic polysaccharide is added as a structural component of a self-healing hydrogel, its chemical modification with introduction of hydrophobic substituents can enlarge the functionalities of the hydrogel; for example, by introducing hemostatic, antibacterial, or adhesive properties.

Hemostatic and antibacterial properties of chitosan, based mainly on the cationic nature of this polysaccharide, are already used for the design of commercially available wound dressing materials [124], but these properties are limited. However, chitosan modification with alkyl side groups significantly enhances these features due to the insertion of the hydrophobes into the membranes of blood cells platelets and bacteria. For example, alkylated chitosan (Ch-Cn) can bind to blood cells and connect them into a network, leading to cell gelation (Figure 6) [125]. Gelation is enhanced by increasing alkyl chain length and polymer concentration, and is reversed by the addition of α-cyclodextrin. Bandages backed with chitosan modified with 4-octadecyl benzaldehyde (DS = 2.5 mol%) were tested on animal injury models and showed a very efficient hemostasis with a 90% reduction in bleeding time in comparison with untreated bandages [126]. Alkylated chitosan also displays significant antibacterial activity when alkyl chain length was C2–C12, both when applied as a solution [127], coated on a gauze [128], or formulated as a sponge [129]. Therefore, alkyl-modified chitosan, mainly Ch-C12, is extensively used for preparation of multifunctional self-healing hydrogels when hemostatic, antibacterial, and adhesive activities are required, especially for application in wound healing. Du et al. [130] prepared a hydrogel via dynamic imine bonds between Ch-C12 (DS = 8.7%) and oxidized dextran (OxD). The gel strength, gelation time, and pore size could be tuned by variation of the ratio of the two modified polysaccharides and the optimal composition was established at Ch-C12/OxD 1/2. The gel self-healing behavior was confirmed both by macroscopic procedure and alternate strain rheological behavior, and the process was fast. Besides, the hydrogel showed high killing efficiency (>95%) against *P. aeruginosa* and *S. aureus* in vitro, good adhesive properties for porcine skin (adhesive strength 3.8–8.9 kPa), much better in vivo hemostatic activity (tested on a rat hemorrhaging liver model) than a similar hydrogel containing unmodified chitosan, and faster wound healing (using a full thickness infected skin defect model) than a traditional gauze.

Further, Du et al. [131] designed a sponge with microchannels created by inclusion of 3D printed polylactic acid microfibers in chitosan solution, freeze-drying the mixture, removal of PLA fiber with dichloromethane and surface modification of chitosan with dodecyl aldehyde (DS = 28% related to chitosan amine groups). A microchanneled structure endowed the sponge with high water and blood adsorption and rapid shape recovery, and the presence of hydrophobic chains determined aggregation of red blood cells and platelets, resulting in a stronger pro-coagulant ability in vitro and a better hemostatic capacity in wound models than clinically used gauzes, as well as efficient antibacterial activity against *S. aureus* and *E. coli*.

Yang et al. [132] prepared a multifunctional hydrogel based on reversible Schiff base linkages between Ch-C12 and PEG dialdehyde, to which tungsten disulfide nanosheets (WS2- NSs) and ciprofloxacin were added, as photothermal therapy agent and antimicrobial agent, respectively. The hydrogel displayed good stability, without significant change in G′ with frequency, a moderate strength (G′ = 90 Pa), self-healing witnessed by step oscillatory strain, and remolding after 3 h. The components of this complex hybrid hydrogel had a synergistic contribution to wound healing. Hemostatic and adhesion properties due to the presence of Ch-C12 were demonstrated by specific tests. Thus, gelation of the heparinized human whole blood took place much faster when in contact with the hydrogel than in the presence of unmodified chitosan and was more efficient than Ch-C12 solution. Its skin tissue adhesion property, evaluated by a lap-shear trial, indicated an adhesive strength of the gel of about 13.7 ± 1.5 kPa, much more efficient than the commercial dressing material Greenplast^®^ (about 5 kPa). WS2-NSs activated by near-infrared light enhanced both ciprofloxacin release and overall antibacterial activity.

Several other reports describe the preparation of hydrogels by cross-linking Ch-C12 with a four-armed benzaldehyde-terminated polyethylene glycol cross-linker [133], or poly-(ethylene glycol) diacrylate [50]. In all cases the hemostatic, antibacterial, and adhesive properties endowed by Ch-C12 led to improved wound healing.

DSs of Ch-C12 used in the preparation of the above described hydrogels are different, but the correlation between this value and hemostatic or antimicrobial performances of the composite was not reported.

## 6. Conclusions

Table 1 summarizes the different types of amphiphilic polysaccharides and their involvement in the design of self-healing hydrogels, this time organized according to the polysaccharide type. There are several conclusions to be drawn from these data.

(1) There are various methods to prepare self-healing hydrogels based on amphiphilic polysaccharides, which provide dynamic hydrophobic associations able to endow the biocompatible materials with desired mechanical and recovering properties.

(2) The wide use of cellulose and chitosan derivatives is obvious, and could be assigned to the wide accessibility of these biopolymers and their intrinsic characteristics. Cellulose and its different derivatives are useful components for hydrogel mechanical properties improvement. Due to the presence of cationic groups, chitosan offers many possibilities for both easy chemical modification and development of materials with new biological activities. There are several other polysaccharides, hyaluronic acid, heparin or pullulan, with known biological activity, which could be very useful in creating multifunctional self-healing hydrogels.

(3) Chemical attachment of low-molecular hydrophobes (C6–C12) is a very facile process, and a very small content of these substituents brings about significant modification in physicochemical properties of the polysaccharides and their applications. For example, attachment of alkyl chains with 4-10 carbon atoms leads to chitosan derivatives with thermogelling properties or to the improvement of its hemostatic, antimicrobial, and adhesive properties.

All these data highlight the potential of amphiphilic polysaccharides and their hydrophobic associations in the design of new, cheap, and efficient biocompatible self-healing hydrogels.

## Figures and Tables

**Figure 1 polymers-15-01065-f001:**
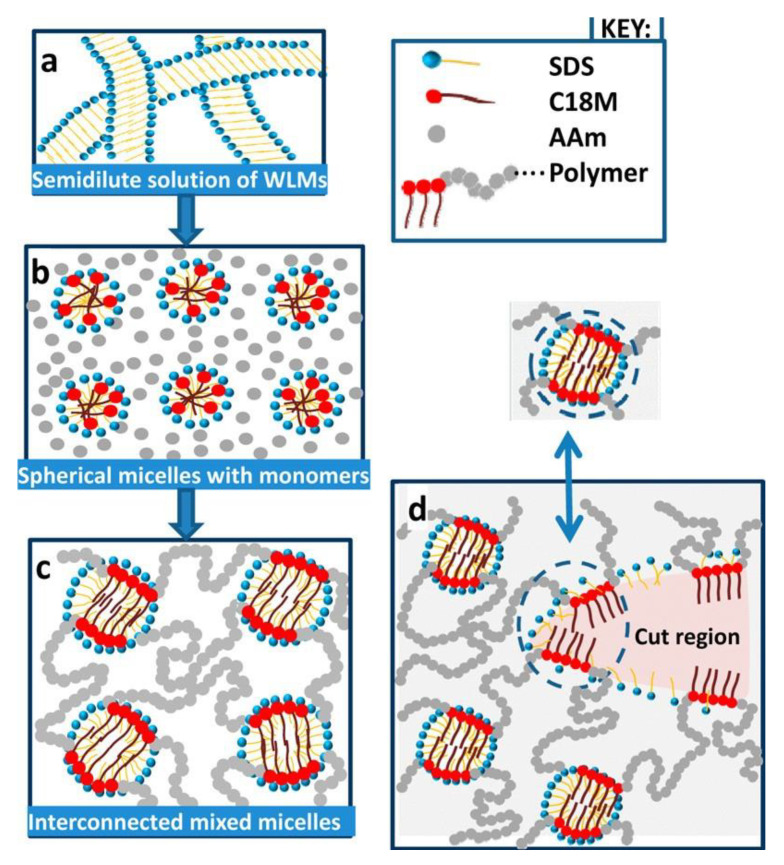
Formation of micellar hydrogels (**a**–**c**) and their self-healing mechanism (**d**). The double arrow in (**d**) indicates the gel region before and after healing. Reprinted with permission from reference [65]. Copyright 2016 American Chemical Society.

**Figure 2 polymers-15-01065-f002:**
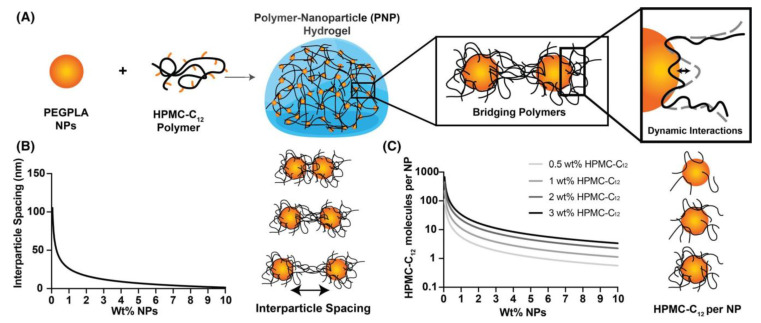
(**A**) Hydrogels formed through the interactions of PEG–PLA nanoparticles (NPs) and dodecyl-modified hydroxypropyl methylcellulose polymers (HPMC-C12). Polymers bridge between polymers and dynamically interact with the NP surface. (**B**) Average interparticle spacing of NPs as a function of the weight percentage of NPs added. (**C**) Number of molecules of HPMC-C12 per NP as a function of the concentration of NPs and concentration of polymer. Reproduced from reference [75], an open access article distributed under the terms of Creative Commons CC BY license, which permits unrestricted use, distribution and reproduction in any medium, provided the original work is properly cited.

**Figure 3 polymers-15-01065-f003:**
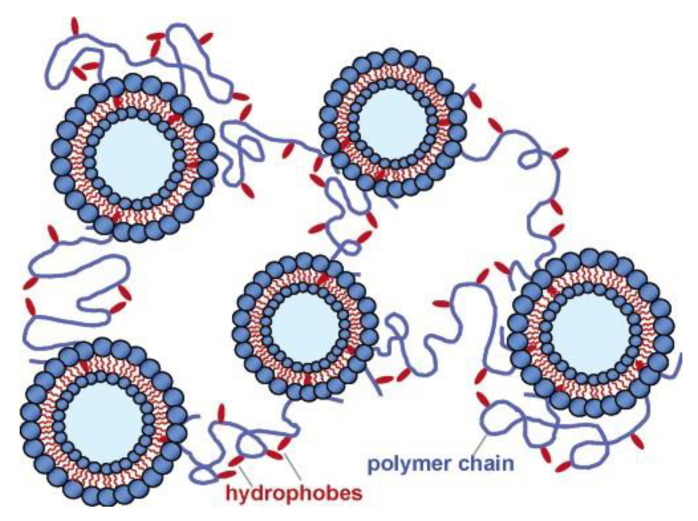
Proposed structure of the network formed upon addition of hydrophobically modified chitosan to vesicles. Polymer hydrophobes are shown to be embedded in vesicle bilayers, thus building a connected network of vesicles. Each vesicle acts as a multifunctional cross-link in the network. Reprinted with permission from reference [71]. Copyright 2005 American Chemical Society.

**Figure 4 polymers-15-01065-f004:**
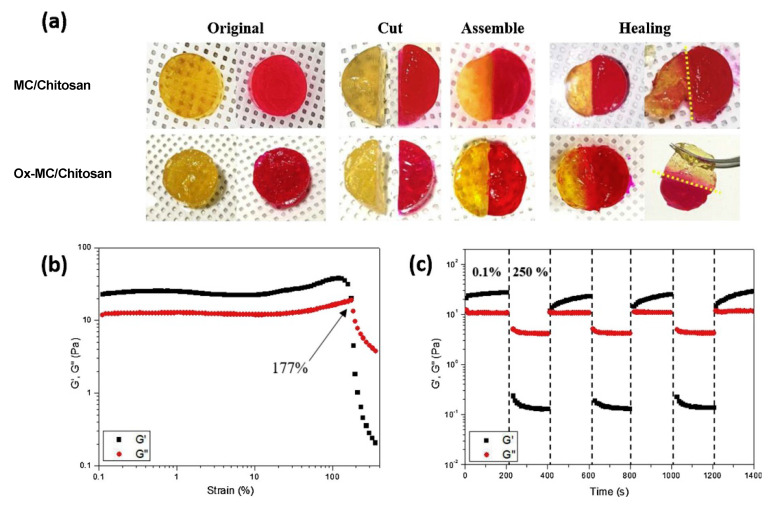
(**a**) Self-healing behaviors of MC/chitosan and Ox-MC/chitosan hydrogels after storage at 37 °C for 20 min. (**b**) Determination of the breaking strain of the Ox-MC/chitosan hydrogel via a strain-sweep test (strain range: 0.1–400%). (**c**) Self-healing behavior of Ox-MC/chitosan via a step-strain test with the application of 250% strain followed by 0.1% strain. All systems contained 8 wt% MC or Ox-MC and 1 wt% chitosan. Reproduced with permission from reference [101]. Copyright 2021 Elsevier.

**Figure 5 polymers-15-01065-f005:**
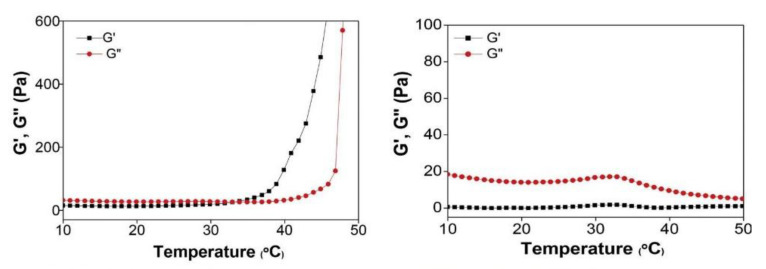
G′ and G″ variation with temperature for GCh-C6 (37%) (**left**) and GCh (**right**), at 4 wt% concentration in water. Reproduced with permission from reference [111]. Copyright 2018 Elsevier.

**Figure 6 polymers-15-01065-f006:**
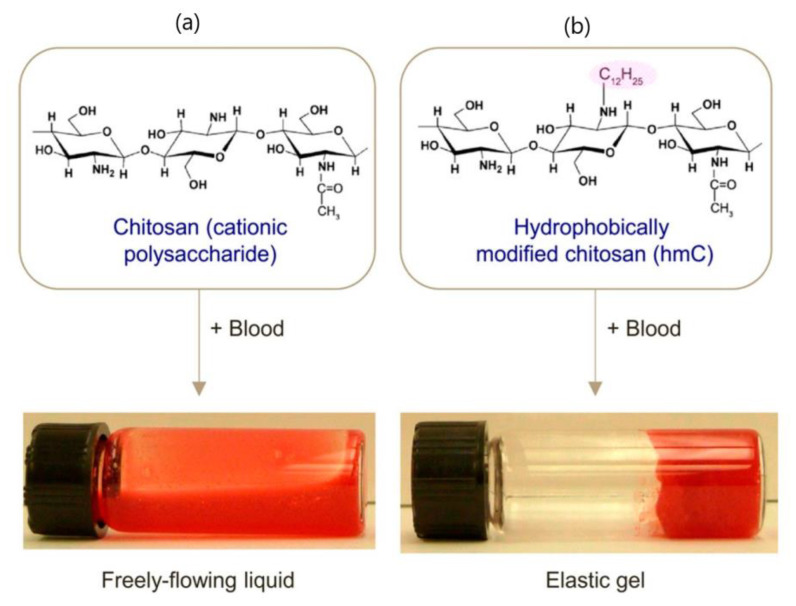
Effect of adding chitosan or hydrophobically modified chitosan (with 5 mol% C12) to bovine blood pretreated with sodium citrate (to prevent clotting), used without dilution. Polymer concentration in both cases was 0.25 wt%. (**a**) Blood remains a flowable liquid when chitosan is added. (**b**) Adding Ch-C12 converts the blood into a gel. The gel has a significant yield stress, which is why it is able to retain its weight in the overturned vial. Reprinted with permission from reference [125]. Copyright 2020 American Chemical Society.

**Table 1 polymers-15-01065-t001:** Amphiphilic polysaccharides and their hydrophobic associations (HA) involved in the preparation of self-healing hydrogels.

Polysaccharide	Derivative	Hydrophobe		Reagents and Synthesis Conditions ^(b)^	Reversible Cross-Links	Applications ^(c)^	Ref.
		Chemical Structure	Content ^(a)^				
Alginic acid	-	Graft C18MeMA	30–70 mM/L reaction medium	Monomer, silk fibroin, water, r.t, 24 h	HA and surfactant	Injectable, biocompatible	[69]
Cellulose	Hydroxypropyl methyl cellulose	Alkyl C12	2%	Alkylisocyanate, TEA, NMP, r.t. 16 h	HA and nanoparticles	Controlled delivery of drugs, vaccines, cells; extensible materials	[72,73,74,75,76,77,78,79,80]
		Alkyl C12 or C18	2–10%	idem	HA and liposomes	Protein controlled release	[85,86]
	Cellulose nanocrystals	Alkyl C8	5%	TCOSi, toluen, r.t. 16 h	Double HA and ionic	Potential biomedical applications	[88]
		Alkyl C8	4 mol%	KIO4, r.t. 48 h octylamine, NaBH_3_CN, r.t. 24 h	HA and Nps, Thermoresponsive	-	[82]
	Carboxymethyl cellulose	diAlkyl (C8)_2_	33 mol%	Dioctylamine, EDC, DMAP, r.t. 24 h	HA	Nontoxic	[60]
	Cellulose nanowhiskers	Graft copolymer Am–C18MeMA	-	Monomers, SDS, APS, TEMED, water, r.t. 24 h	HA and surfactant	-	[66]
	Methylcellulose	-	-	-	Thermoresponsive HA, imine bonds	Controlled drug delivery	[101]
Chitosan	-	Grafted PU	15–30%	PU-isocyanate end groups, DBTDL, acetic acid/DMF, 80 °C	HA	Injectable, sustained drug delivery	[62]
		Ferrocene	8.2 mol%	Fc-COOH, EDC, NHS, r.t. 48 h	HA	Drug delivery	[61]
		Alkyl C12	2.5–10 mol%	n-Dodecyl aldehyde NaBH_3_CN, r.t., water/EtOH, 24 h	HA and vesicles	-	[71,83,84]
			8–28%	idem	Imine bonds	Wound dressing with hemostatic, antimicrobial and adhesive properties	[130,131,132,133]
		Hydroxybutyl	1.3–1.7/r.u.	EBu NaOH 50%, 60 °C or KOH/urea. r.t.	Thermoresponsive	Wound dressing, tissue engineering, cell culture, cell and drug carrier	[102,103,104,105,106,107,108,109]
	Glycol chitosan	ABA copolymer with benz aldehyde side groups	0.05–0.32 molar ratio CHO/NH_2_	ABA with CHO groups Water pH 7.4, r.t.	HA and imine bonds	Potential biomedical applications	[90]
		Alkyl C6	37%	HexAnh, water/MeOH r.t. 48 h	Thermoresponsive	Tissue engineering, ocular delivery, wound healing	[110,111,112,113]
	Carboxymethyl chitosan	Alkyl C6	0.5 mol/r.u.	-	Ionic	Micelles loaded with drug	[121,122]
Dextran	Carboxymethyl dextran	Grafted PCL	5%	PCL wit OH end groups, EDC, DMAP, DMSO, 37 °C 48 h	HA and irreversible covalent bonds	Tissue engineering	[89]
	-	Alkylamide	>1 mol/r.u.	Amic acids, DCC, DMAP, DMSO, 18 h	HA	Injectable as colloidal systems	[115]
Dextrin	- -	Grafted PU	-	PU-isocyanate end groups, DBTDL DMF, 70 °C	HA and surfactant	Injectable, local drug delivery to tumor site	[63]
		Alkyl C16	8–14 mol%	Palmitic acid, CDI, DMSO, 80 °C, 5 h	Imine bonds	Micelles loaded with drug	[123]
Galactomannan	-	Graft copolymer Am–C18MeMA	-	Monomers, SDS, APS, TEMED, water, r.t. 24 h	HA and surfactant	-	[68]
Hyaluronic acid	-	Alkyl Cn, n = 8, 12, 14	-	TBA-OH, alkylamine, DCC, DMSO, r.t. 24 h	HA and NPs	Catheter-injectable hydrogel	[81]

^(a)^ Content as reported in the cited references; the unities (mol%, mol/repeating unit (r.u.), wt%, number/chain) are not always provided. ^(b)^ Synthesis of the amphiphilic polysaccharide. ^(c)^ According to in vitro/in vivo tests reported.

## Data Availability

No applicable.

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
