# Peer review of "Role of Hydrophobic Associations in Self-Healing Hydrogels Based on Amphiphilic Polysaccharides"

_polymers, 2023, doi:10.3390/polym15051065_

Round 1

Reviewer 1 Report

I enjoyed this paper. The importance of this paper is high enough to warrant publication in Polymers,

Author Response

Thanks to reviewer for appreciation.

Reviewer 2 Report

Marieta Nichifor did a perfect work preparing this manuscript. I would especially like to note two aspects of the manuscript. The first is the brief theoretical excerpts describing the fundamentals of hydrophobic interactions and their applications to the production of hydrogels. The second is Table 1 summarizing of a large part of the cited papers. An excellent work that can be useful not only for scientists, but also for those who are just starting their research path.

I would like to thank the author for such an excellent work. I can only recommend to add a list of abbreviations, because sometimes it was difficult to navigate in such a voluminous information array.

Author Response

A list of abbreviations was placed at the end of manuscript, before references. The list does not include the abbreviations used only in the framework of a single paragraph.

The text was carefully checked for spelling error corrections.

Reviewer 3 Report

The review paper for consideration tackles self-healing character of polysaccharide-based hydrogels formed from hydrophobic associations. The paper overall is satisfactorily written to cover the target topic. I have the following comments and/or suggestions to the author - these are as follows:

(1) Polysaccharide-based self healing hydrogels can be prepared from various methods such as hydrophobic interaction, ionic crosslinking hydrogels, in situ polymerization or dynamic covalent bond-based interaction. What is so special about hydrophobic-based interaction hydrogels using polysaccharides over types of hydrogels prepared by other methods such as mentioned above? This should be highlighted and given a priority in terms of giving an in-depth discussion in  the early part of the manuscript.

(2) Include a tabular list of references to include in different columns to include type of polysaccharide used to make the hydrogel; reaction condition/modification methods; type of hydrophobic moiety, highlighted properties and/or possible application; references.

(3) The discussion on the Evaluation of self-healing hydrogel performances can be improved by, while introducing different techniques to characterize self healing properties, the papers used in this part/chapter are polysaccharide-based hydrogels formed on hydrophobic associations.

I recommend publication with revisions, as pointed above.
